# Structure, Function, and Allosteric Regulation of the 20S Proteasome by the 11S/PA28 Family of Proteasome Activators

**DOI:** 10.3390/biom13091326

**Published:** 2023-08-29

**Authors:** Taylor Thomas, David Salcedo-Tacuma, David M. Smith

**Affiliations:** 1Department of Biochemistry and Molecular Medicine, School of Medicine, West Virginia University, 64 Medical Center Drive, Morgantown, WV 26506, USA; 2Department of Neuroscience, Rockefeller Neuroscience Institute, West Virginia University, Morgantown, WV 26506, USA

**Keywords:** proteasome, protein degradation, proteostasis, 11S, REGs, PA28γ

## Abstract

The proteasome, a complex multi-catalytic protease machinery, orchestrates the protein degradation essential for maintaining cellular homeostasis, and its dysregulation also underlies many different types of diseases. Its function is regulated by many different mechanisms that encompass various factors such as proteasome activators (PAs), adaptor proteins, and post-translational modifications. This review highlights the unique characteristics of proteasomal regulation through the lens of a distinct family of regulators, the 11S, REGs, or PA26/PA28. This ATP-independent family, spanning from amoebas to mammals, exhibits a common architectural structure; yet, their cellular biology and criteria for protein degradation remain mostly elusive. We delve into their evolution and cellular biology, and contrast their structure and function comprehensively, emphasizing the unanswered questions regarding their regulatory mechanisms and broader roles in proteostasis. A deeper understanding of these processes will illuminate the roles of this regulatory family in biology and disease, thus contributing to the advancement of therapeutic strategies.

## 1. Introduction

The proteasome is a multi-catalytic protease complex that functions as a master regulator of cellular homeostasis mechanisms through the efficient degradation of soluble proteins [1,2,3]. From the moment the proteasome was discovered, immense effort has been expended to decode its role as a protease and to elucidate the diverse mechanisms that govern proteolysis. These regulatory mechanisms, including highly specialized proteasomal activators, adaptor proteins, post-translational modifications (PTMs), and others, create an intricated network for the maintenance of proteostasis and cellular function. The proteasome has been shown to play a role in virtually all cellular processes, e.g., cell cycle progression [4,5,6], DNA repair [7,8], regulation of gene expression [9], apoptosis [10], immunological responses [11], and disease pathogenesis [12,13]. The functional role of the proteasome remains unchanged in most of these pathways (degradation of soluble proteins). However, specific proteasome functions can significantly differ in different cellular settings involving unique regulatory components. A general classification of proteasomal regulatory mechanisms can be based on the regulatory component’s ability or inability to unfold protein substrates. The purpose of this review is to introduce general aspects of proteasomal regulators and specifically discuss the evolution, cell biological roles, and allosteric regulatory mechanisms of the mammalian proteasome activator 28 (PA28) family.

## 2. An Overview of Proteasome Regulation

### 2.1. Unfolding-Dependent Protein Degradation

Protein degradation occurs in the 20S (core particle) proteasome, which is a barrel-like structure (Figure 1) that harbors protease sites on its interior. For proteins to access the interior degradation chamber they must be unfolded to be able to first pass through the proteasome’s narrow entry pore [14]. Generally, this process requires an unfoldase, using a forced threading mechanism to unfold the soluble protein substrate and immediately inject it into the 20S for degradation. Two eukaryotic proteasomal regulators are classified as unfoldases: the 19S regulatory particle (RP) and p97 (Yeast: CDC48 [15]/Archaea: VAT [16] (Figure 1). Both the 19S RP and P97 belong to the AAA+ ATPase (ATPases are associated with various cellular activities) family, and maintain the ability to unfold substrates, and translocate them into the degradation chamber of the proteasome. The 19S RP is a 700 kDa macromolecular complex that binds to the 20S core particle (20S CP) and forms the 2.5 MDa 26S proteasome complex [17,18]. The 19S is a highly specialized proteasomal regulator as it can participate in both ubiquitin-dependent and -independent protein degradation. Ubiquitin-dependent protein degradation requires the cooperation of the ubiquitin-conjugation system (e.g., E1, E2, and E3 enzymes reviewed here [19], which selects proteins for ubiquitination to target them to the 26S for degradation. This degradation system is known as the ubiquitin–proteasome system (UPS). The UPS catalyzes the elimination of damaged, misfolded, or functionally unnecessary proteins and is responsible for the bulk of total intracellular protein degradation. Similarly, p97 recognizes ubiquitinated proteins and can catalyze their degradation by the proteasome. However, while the P97–20S complex has been shown in archaea, its existence in eukaryotes is not directly established. P97 enlists a complex of two proteins, Ufd1 and Npl4 (UN complex), that bind to ubiquitin chains to recruit substrates [20] and plays an important role in endoplasmic reticulum-associated degradation (ERAD) [21]. However, ubiquitin is not always required for protein degradation; the 26S proteasome has been demonstrated to participate in Ub-independent protein degradation [22,23,24]. In addition, other Ub-independent protein degradation mechanisms exist and require alternative proteasomal regulatory complexes. However, these other regulatory complexes do not have the ability to unfold their substrates on their own.

### 2.2. Degradation of Unstructured Proteins (Ubiquitin-Independent)

Unstructured proteins, or proteins with high intrinsic disorders (IDPs), can be degraded by several proteasome complexes in ubiquitin-independent fashions. The 20S by itself, and the PA200–20S and PA28–20S complexes, can directly catalyze the degradation of unstructured proteins. These degradation pathways are particularly important to health and disease as IDPs are most often found associated with disease, especially cancer and neurodegenerative diseases [25,26,27].

20S-dependent protein degradation: A lot of work has expanded on the regulatory mechanisms surrounding the ability of 20S CP to degrade and process proteins in the absence of a proteasomal regulator. The extent that this pathway is used in the cell is unclear, but it is interesting that the 20S core particle is the most abundant species in most cell types [12], and its levels are upregulated in stressful conditions, where other proteasome complexes (e.g., 26S) are disassembled [28,29]. For example, the 20S is upregulated by periods of oxidative stress in the cell. Oxidative stress is known to cause damage to soluble proteins in the form of solvent-exposed hydrophobic patches and protein unfolding, which can lead to protein degradation [30], protecting the cell from deleterious protein aggregation [31]. The 20S has also been implicated in the degradation of IDPs generally, which is nicely reviewed in detail by the Glickman group [12,14].

PA200/Pi31: A variety of Ub-independent proteasomal regulators have been identified and characterized with their own unique regulatory mechanisms, cellular localizations, and physiologic functions that diversify proteasomal degradation. These regulatory complexes each function with the 20S CP in a unique way and are not known to have the intrinsic ability to unfold their protein substrates. PA200 (Yeast: Blm10) is a nuclear proteasomal regulator that can target acetylated histones for degradation during DNA damage repair, spermatogenesis, and chromatin remodeling [32]. A recent structural study on the PA200–20S CP complex revealed that PA200 induces structural rearrangements in the 20S CP that increase substrate entry and change the functional capabilities of the 20S CP protease active sites [33]. It has also been shown to form a ternary hybrid complex with the 26S proteasome (19S RP–20S CP–PA200), which could provide unique functions to its role in physiology [34]. Proteasome Inhibitor 31 (PI31) is a proteasome adaptor protein that was discovered as a homodimer that interestingly inhibits proteasomal activation induced by PA28 [35]. More recently, PI31 has been shown to bind the 20S CP and facilitate proteasome assembly and transport in neurons [36]. Its cryo-EM structure has now been partially solved in the bound state with the 20S. These structures provide a mechanism for how PI31 impairs the proteolytic sites of the 20S [37]. The general cell biological roles of these two proteasome regulatory complexes have not yet been made clear.

11S/PA28 Family: The PA28 Family, also known as 11S regulators, constitutes a fascinating member of ubiquitin-independent proteasomal regulation. This family comprises heptameric, ubiquitin-independent proteasomal regulators, and in the domain of higher eukarya, it is represented by three homologs: PA28α (PSME1), PA28β (PSME2), and PA28γ (PSME3). Intriguingly, PA28α and PA28β form mixed heteroheptameric complexes, commonly known as PA28αβ, under natural physiological circumstances, yet they can form homohexamers when individually expressed in bacteria [38]. On the other hand, PA28γ exists physiologically as a homoheptamer [39]. The 11S family also includes the homoheptameric complex PA26 from *Trypanosomes*, which was the first 11S regulator to be structurally characterized in complex with the 20S CP [40,41]. Despite having large sequence and structural variation between PA26 and the PA28 homologs, the proteasomal regulatory mechanisms are highly conserved between PA26 and PA28αβ (they similarly induce 20S gate-opening), but PA28γ can be considered a functional outlier [42]. 

PA28αβ expression is induced by interferon-γ (INF-γ), and the regulatory complex localizes to the cytosol. Functionally, PA28αβ has been demonstrated to form a complex with a specialized 20S CP for the immune system, the immunoproteasome, to aid MHC Class I antigen presentation. PA28αβ demonstrates the capacity to accelerate the entry of unfolded substrate or the accelerated exit of peptides from the 20S CP affecting their size. This is hypothesized to stimulate the production of a broad spectrum of peptides suitable for antigen presentation [43,44]. Furthermore, recent efforts reveal that during periods of oxidative stress, such as those encountered in tumors, the expression of PA28αβ is enhanced. This expression is thought to promote the breakdown of oxidatively damaged proteins through a process that operates independently of ubiquitin and ATP [43].

Conversely, PA28γ is predominantly located in the nucleus and is expressed constitutively. It has also been hypothesized to regulate the 20S CP through mechanisms distinct from those of PA28αβ. As opposed to controlling substrate entry, PA28γ has been suggested to enhance the proteolytic active site of the 20S CP, which is responsible for cleavage after basic amino acids [39], a function recently demonstrated conclusively by Thomas and Smith in 2022 [45]. Enigmatically, alterations such as a single-point mutation and changes in purification strategies have been found to shift PA28γ’s function from a trypsin-like activator to a gate-opener [44,46,47]. In a physiological context, PA28γ has been shown to promote the degradation of unstructured proteins via the 20S CP [48], though the mechanism is yet to be fully elucidated. Gaining insights into these biochemical mechanisms is crucial, as PA28γ expression has been associated with certain disease states, a topic detailed thoroughly in the discussion of Stadtmueller and Hill, 2011 [42]. Despite the literature on mammalian PA28 family regulators and their links to different roles across cellular/physiologic processes, several fundamental questions still exist regarding their inherent proteasomal regulatory mechanisms and their broader roles in proteostasis.

It remains uncertain how these regulators select substrates, or even stimulate protein degradation in vivo. Particularly for PA28γ, it is yet undetermined which classes of proteins serve as substrates for these activators, as well as the methods they employ to manage substrate selection and stimulate entry into the 20S CP for degradation. While the physiologic role of PA28αβ in MHC Class I antigen presentation appears to be clear, PA28γ’s biochemical and physiologic role in proteostasis remains ambiguous. This review concisely combines and presents the current PA28 literature and draws attention to critical questions to fully understand this proteasomal regulatory family, including its functions in biology and disease.

## 3. Proteasome Structure and Function

Since the PA28 family binds to and regulates the 20S CP, it is important to understand the biochemical and structural components of the 20S. Briefly, the proteasome, more specifically the 20S CP, is a hollow, barrel-shaped, and compartmentalized protease. The eukaryotic 20S CP contains four heteroheptameric rings, two α-rings, and two-β rings, in an α, β, β, α arrangement [49]. Each of the α-ring’s seven subunits features N-termini with the ability to assume one of two regulated conformations—open or closed—and thus governing substrate entry/exit [49,50]. This arrangement is frequently termed as the proteasome’s “gate”. Strong hydrophobic interactions between the α- and β-rings prohibit substrates from entering the degradation chamber other than through an opened gate positioned at either end of the 20S CP [51], thus preventing non-specific protein degradation. The protein degradation function is performed through the catalytic subunits located in the β-rings. In each β-ring, three N-terminal threonine protease subunits are present (six catalytic subunits in total), all aligned towards the hollow center of the proteasomal degradation chamber [49]. Each proteolytic subunit exhibits a unique specificity for certain peptide chemistries (Figure 2). The β1 subunit cleaves peptides following acidic amino acids (also known as the caspase-like site, or CP-L) [52]. β2 operates by cutting after basic residues (termed the trypsin-like site, or T-L), while β5 cleaves peptides after hydrophobic residues (referred to as the chymotrypsin-like or CT-L site) [53]. In tandem with its inherent regulatory mechanisms, the diverse set of catalytic subunits enable the 20S CP to be a supremely controlled protein degradation machine.

The biological imperative of maintaining a closed gate to prevent non-specific protein degradation reveals the requirement for proteasomal regulators to ensure selective substrate entry to induce gate-opening and proper proteolysis for only recruited substrate entry. To facilitate proteasomal regulator binding, the α-rings possess intersubunit pockets, which are used to promote the docking of various proteasome regulatory complexes, thereby promoting specific substrate entry. Specifically, proteasomal AAA+ ATPases, PA200, and the 11S regulators have been found to dock into the intersubunit pockets of the 20S CP α-rings, prompting gate-opening in diverse ways [33,40,54]. For example, proteasomal AAA+ ATPases require ATP, or non-hydrolysable ATP analogs, to associate with the 20S CP, a prerequisite not needed for the 11S regulators [39,54]. PA28αβ and PA26 have been shown to induce gate-opening when they bind to the 20S CP, discussed in depth below [40,41]. In terms of functionality, the PA28αβ–20S CP gate-opening mechanism might enhance the MHC Class I antigen presentation [55,56,57]. In parallel to this role for PA28αβ in antigen presentation the immunoproteasome, a specialized 20S complex, plays similar roles. During the immune responses, the standard catalytic subunits of the 20S proteasome, specifically β1, β2, and β5, are substituted by the immuno-subunits β1i/LMP2, β2i/MECL-1, and β5i/LMP7, respectively [58,59,60]. This reconfiguration of the 20S protease sites enhances the efficiency of antigen processing, resulting in peptides that are better suited for binding to MHC Class I molecules, thereby ensuring a more refined immune recognition and response [59,61]. The specialized nature of the immunoproteasome, characterized by these specific subunit substitutions, highlights the proteasome’s adaptability to fulfill the dynamic requirements of the immune surveillance. On the other hand, the regulatory strategy of the constitutively expressed PA28γ is perplexing to say the least. Remarkably, the literature reports contradicting mechanisms of either gate-opening or catalytic site activation [39,47]. Both mechanisms hold substantial supporting evidence, leading to subsequent studies seeking to either validate or refute the other. However, it is plausible to consider that these are not two conflicting regulatory mechanisms, but rather perhaps both PA28γ mechanisms regulate the proteasome in different contexts (more below).

## 4. Diversity in the 11S Family of Proteasomal Regulators

The 11S family of regulators are prevalent among the kingdoms in Eukarya. They are found as early as unicellular eukaryotes, e.g., PA26 from *Trypanosoma brucei* [62] and PA28 homolog *Plasmodium falciparum* (PA28*Pf*) [63]. PA28α, PA28β, and PA28γ are found throughout the animalia kingdom. The presence of diverse 11S regulators throughout Eukarya poses a unique question about the evolutionary/historical function of proteasomal regulators [64]. 

A preliminary analysis using PA28γ sequences from representative organisms in the NCBI database reveals a conserved profile throughout the evolutionary history of vertebrates (indicated by the orange shade in Figure 3). Interestingly, at some point during evolution this γ variant appears to have diverged into α and β forms in higher animals, presumably in response to the evolving complexity of the 20S proteasome (immunoproteasome) and the adaptive immune system. In contrast, in invertebrates and simpler organisms the γ form prevails, potentially due to a less complex 20S proteasome [64]. Specifically, PA28 from *Plasmodium falciparum* (PA28Pf), *Drosophila melanogaster* REG gamma (fruit fly), nematodes *(Caenorhabditis elegans*), and corals (*Acropora millepora*) appear to share more features with the PA28 α and β variants than with the vertebrate PA28γ, forming a distinct cluster (Figure 3) (denoted by the blue shade) [65,66]. This suggests two major groupings within PA28γ regulator family: a highly conserved PA28γ variant among vertebrates and a second cluster of PA28/REG gamma orthologues in non-vertebrates that align more closely with the α and β variants of higher vertebrates.

It is tempting to speculate that this early form of PA28 might represent an ancestral version that gave rise to the α, β, and γ variants seen in vertebrates, though further analysis would be needed to support this [64,65,66]. Unique to mammals are the PA28α and PA28β variants, which play roles in the MHC Class I antigen presentation pathways typically induced by INF-γ or adaptive immune responses: responses that originated in vertebrates [30]. This observation could suggest that these early “invertebrates” variants might influence similar PA28α/β functions in the immune systems of the organisms that possess them in unique ways. However, a more comprehensive analysis is required to understand the details of the phylogeny and functional impact associated with this specific family of regulators. Considering the complexity of the PA28 gamma evolutionary history plus the vast body of research on PA28α and PA28β, we still do not know why PA28γ remains underexplored, especially given its significant potential.

Despite our incomplete understanding of these regulators and their evolutionary roles throughout Eukarya, insights from structural and computational biology have proven invaluable in understanding the structural similarities and distinctions among 11S regulators. A hallmark feature conserved in all 11S regulators is their heptameric tertiary structure, with each subunit embodying four parallel alpha helices (Figure 4A). In addition, an invariant 11S motif known as the activation loop resides in the linker region between helices 2 and 3 (Figure 4B) [44]. Although this activation loop sequence is similar between PA26 and PA28, it is almost identical within the higher eukaryotes PA28α, PA28β, and PA28γ. The activation loop is critical for proteasomal activation by PA28αβ, PA28γ, and PA26, as mutagenesis in this region results in a loss of the ability to activate gate-opening in the 20S CP or even form stable heptamers [67].

The region linking helices 1 and 2 in the 11S regulators presents significant divergence. PA26 is notably shorter compared to its counterparts in the higher eukaryotes PA28 family. The PA28 family features a linker region motif known as the homolog-specific insert. This insert is an intrinsically disordered region (IDR) varying in length and chemical composition between PA28α, PA28β, and PA28γ, and will be referred to as the IDR in this review (Figure 4C). The position of the IDR motif places it as surrounding the putative substrate entry pore of the 11S members; thus, it is in a good position to interact with substrates prior to degradation. The IDR of PA28α presents a repeating lysine (K) and glutamate (E) motif, called the KEKE motif [68]. PA28β, which is recognized for hetero-heptamerizing with PA28α, exhibits similar chemistry, and is possibly essential for the formation of the PA28αβ complex [67]. The IDR of PA28γ, on the other hand, is the longest and has limited chemical similarity to the IDRs of PA28α or PA28β. PA28γ’s IDR comprises a cluster of positive residues followed by a cluster of negative residues (Figure 4C). Interestingly, the presence of the IDR does not influence the PA28γ complex assembly nor its functional interactions with the proteasome [45,68]. The structural characterization of these IDRs, either individually through NMR or within the context of the protein complex, remains elusive, leaving the question of their physiological roles open-ended.

Interestingly, PA26 features a unique internal pore loop within the substrate channel located above the activation loops, a feature not identified in the PA28s of higher eukaryotes (Figure 4A(iii), purple). Other regions with low sequence similarity among the 11S regulators include the residues in the substrate-entry pore, N-termini, C-termini, and those in proximity of the activation loop [44]. Considering these observations, it seems plausible that the variation in IDR length, low sequence similarities, and the presence of additional structural motifs contribute to the diversity of functional responses amongst the 11S regulators. This is particularly true when examining the mammalian PA28αβ and PA28γ in complex with the 20 CP proteasome. We will focus on the structural and functional differences between the mammalian PA28αβ and PA28γ in detail below. 

## 5. PA28αβ

### 5.1. Structure-Function

PA28αβ has been shown biochemically and structurally to induce 20S CP gate-opening using a mechanism similar to PA26. To initiate 20S CP gate-opening, PA28αβ and PA26′s C-termini provide the binding energy necessary to dock into the intersubunit pockets of the proteasome’s α-ring and form a stable quaternary structure. Conserved amongst the 11S family members’ C-termini is a terminal tyrosine (Y) residue and a proline (P) in the -10 position that confers the energy necessary for binding, but the chemical identity amongst the other eight residues varies [69]. The unstructured 11S C-termini form structured helices upon binding in the 20S CP intersubunit pockets; this ordered structure adequately positions the 11S regulator’s C-terminal carboxy group to form a salt bridge with a specific lysine in the intersubunit pocket of the α-subunits, and provides the stability for the activation of loop contact and function in the PA28–20S interface and its downstream causal mechanism of 20S CP gate-opening [41]. Mutagenesis of the terminal residue(s) of PA28α, PA28β, or PA28γ’s C-termini abolished proteasome binding, demonstrating the importance of C-terminal binding [69,70]. 

After the 11S–20S CP complex is stabilized by C-terminal docking, the reverse-turn of the activation loop is ideally in position to activate the 20S CP. However, as PA28αβ and PA26 are heteromeric and homomeric, respectively, there is some variability in their ability to open the gate. PA26 has perfect seven-fold symmetry of its activation loops due to its seven identical subunits, which PA28αβ does not; in humans, recent research showed PA28 stoichiometry of 3α/4β, which is different from the mice stoichiometry 4α/3β reported, as the preferred conformation to engage the 20S proteasome [71]. Therefore, PA26 can fully open the 20S CP gate, whereas PA28αβ only partially opens the gate due to asymmetrically binding to the 20S CP [71,72]. Nevertheless, the gate-opening mechanism of these 11S regulators remains the same, just to different extents. PA28αβ and PA26 are positioned to create contact forces against the pseudo-seven-fold symmetry of the mammalian 20S CP’s N-terminal α-subunits reverse-turn loops (defined by Pro17) and cause the α-subunits to become more symmetric and less-likely to be in an asymmetric closed-gate conformation [41]. This open-gate conformation is quantifiable in a proteasome activity assay using fluorogenic peptide substrates that are specific for each of the three proteolytic subunits. When compared against a wild-type proteasome, PA28αβ–20S CP complexes can enhance 20S peptide degradation rates by 200-fold [39,45], which is the result of inducing 20S gate-opening. What is also important about this gate-opening function is the ability of the PA28αβ–20S CP complex to form a hybrid complex with the 19S RP. Using negative-stain transmission electron microscopy (TEM), the 20S CP was visualized to be capped on one end by PA28αβ and the other by the 19S RP in a 19S RP–20S CP–PA28αβ assembly [73]. It is hypothesized that this hybrid complex is used for PA28αβ’s physiologic function in the major histocompatibility complex (MHC) Class I antigen presentation. While the function of the CP–PA28αβ complex in the immune system is still not completely clear, in vitro studies suggest that it increases the diversity of the larger sized peptides that are compatible with the MHC Class I presentation [56,57].

### 5.2. Physiology

PA28αβ expression, along with specific 20S CP β-catalytic subunits (β5i/LMP7, β2i/MECL-1, and β1i/LMP2) that substitute the constitutive subunits to form the immuno-20S CP (i20S CP), is driven by the cytokine INFγ. The PA28αβ–i20S CP complex has been demonstrated to increase the hydrophobicity and average length of peptide epitopes (approximately 8–10 amino acids long), optimizing them for binding to the MHC Class I heterodimers and presenting them at the cell’s surface [56,74]. In addition, PA28αβ can increase the type of peptides generated, providing a greater repertoire of peptides for MHC receptors, thus increasing the efficiency of presented peptides in immunity challenges [56,57,75]. Under normal conditions, the peptide epitope comes from the cell’s pool of expressed proteins and the immune system is nonreactive to these “self” antigens. However, under diseases (e.g., cancer) or infection from a viral or bacterial source, these MHC Class I molecules display pathogenic, or “foreign”, peptides, which trigger the immune response and terminate the cell via cytotoxic T lymphocytes [76]. Even with the current knowledge on PA28αβ’s function, research is ongoing to conclusively establish its importance and role in immune function [77]. 

Recently, the role of PA28αβ has been spotlighted, especially within the context of oxidative stress. This regulator has been explored as a component of the pathway of degradation for protein substrates damaged by oxidative stress [78,79,80]. Notably, during oxidative challenges PA28αβ, in conjunction with the i20S CP, demonstrates a strong induction: up to three-fold during oxidative stress. Oxidatively damaged proteins, under these circumstances, seem to be targeted and degraded by the PA28αβ–i20S and 20S CP complex [30,78,79]. The capacity of cells to break down oxidatively damaged proteins rises in tandem with this strong induction of 20S and PA28αβ synthesis. Moreover, knockdown treatments of 20S proteasome, immunoproteasome, and PA28 result in a significant reduction in this degradative capacity. Particularly, PA28αβ-knockout mutants could only achieve half of the H_2_O_2_-induced adaptive increase in proteolytic potential compared to wild-type controls [78,79]. These insights open the door for more detailed research into the diverse and precise functions of the 11S complexes that serve in both proteostasis and immunology.

## 6. PA28γ

### 6.1. Structure-Function

Unlike PA28αβ, the precise biochemical and physiological mechanisms of PA28γ remain largely unclear. This is due, in part, to the fact that there has been a scarcity of structural information reported about PA28γ or the PA28γ–20S CP complex until recently. Since PA28γ’s discovery in 1990 from systemic lupus erythematosus (SLE) patient serum, its functional role with the 20S CP has been heavily debated [81]. It was initially reported that PA28γ was an activator of peptide substrate hydrolysis targeted only for the trypsin-like (T-L) 20S β-catalytic site, and further downregulated the chymotrypsin-like (CT-L) and caspase-like (CP-L) β-catalytic sites [39,44]. Another group proposed that PA28γ was not peptide substrate-specific and that it was able to degrade peptide substrates for all three β-catalytic sites [47], suggesting a 20S gating function. Interestingly, a single-point mutation in PA28γ from lysine 188 to glutamate (K188E) seemed to demonstrate a shift in 20S CP regulation from T-L β-catalytic activation to gate-opening [44]. Ultimately, different experimental protein purification strategies were the explanations for the different 20S CP regulatory mechanisms, as overexpressed FLAG-PA28γ purified from COS7 cells was suggested to be T-L-activating [82]. Even though PA28γ has been characterized as a T-L activator of the proteasome, this finding yields numerous questions about how PA28γ explicitly increases T-L activation while other 11S family members function primarily to open the 20S gate, showing no specificity for peptide substrates. The most crucial question awaiting an answer is the process by which PA28γ activates the 20S CP. Understanding this mechanism would significantly increase our understanding of the literature surrounding PA28γ’s physiological role, which we will discuss in detail below.

PA28γ could use any one of the following biochemical mechanisms to implement its role as a T-L activator: (1) allosteric activation of the T-L 20S CP β-catalytic site, (2) a smart filter selective for T-L β-catalytic site peptide substrates, 3) inhibition of the CT-L or CP-L β-catalytic sites, (4) conversion of the CT-L or CP-L β-catalytic sites to T-L-activating, or (5) 20S CP gate-opening in combination with one of the aforementioned mechanisms. These hypothetical mechanisms for PA28γ are reasonable as examples of such mechanisms have been observed in other proteasome regulators. For instance, proteasomal ATPases utilize allostery to confer the conformational changes necessary to open the 20S CP and translocate target protein substrates into the degradation chamber [54,83,84]. PA28αβ has been suggested to act as a “smart sieve” to produce peptides of the appropriate length for MHC Class I antigen presentation [43,56]. In addition, previous biochemical reports using affinity labeling indicated that PA28γ inhibited the CT-L and CP-L β-catalytic sites [44]. Some other examples from PA200, a fellow nuclear and T-L-activating proteasomal regulator, showed changes in the S1 side pocket chemistry of the CT-L β-catalytic site from hydrophobic to acidic, which changes β5 from CT-L to T-L activity [33]. In general, 11S regulators PA26/PA28αβ/PA28*Pf* have all been structurally determined to open the gate of the 20S CP to different extents [41,63,72]. Therefore, to understand how PA28γ functions in proteostasis and disease it is crucial to understand how PA28γ regulates 20S CP activity.

Recently, Thomas and Smith established PA28γ’s function as a T-L proteolytic-site activator, using allosteric regulation to activate the 20S CP in the presence of peptide substrates without requiring any supplementary mechanisms to fulfill this role [45]. The study used a constitutively open 20S CP to deconvolute gateing contributions to the quantified enzymatic activities. This approach disambiguated the potential mechanisms of activation and determined that PA28γ could allosterically activate the 20S T-L peptidase active site in the 20S β-ring. An illustrative representation of the allosteric effects of PA28αβ on the 20S is shown in Figure 5E. This study also presented the first cryo-EM density map of the PA28γ–20S CP complex, exhibiting a quaternary structure similar to other 11S–20S CP complexes (Figure 5A). The map showed that PA28γ binds with the 20S CP through its C-termini into the intersubunit pockets of the 20S CP α-rings. Upon alignment with the PA28αβ–i20S CP complex model, a remarkable similarity in secondary and tertiary structures between PA28γ and PA28αβ was observed (Figure 5B,C). Despite their secondary and tertiary structure similarities, a hydrogen–deuterium exchange coupled with mass spectrometry (HDX–MS) study performed on PA28αβ and PA28γ alone and in complex with the 20S CP and i20S CP revealed that both 11S regulators have unique solvent exposures in complex with the 20S CP, likely due to differing dynamics (Lesne et al., 2020). Subsequent to this initial publication of the PA28γ–20S complex, Chen, 2022, produced a 3.4 Å structure of PA28y [85], adding further support to the structure and similarities discovered by Thomas and Smith earlier that year (Figure 5D). While high-resolution structural analysis is needed to precisely define the molecular mechanism by which PA28γ enhances 20S T-L activity, existing structural studies suggest that despite structural resemblances between PA28αβ and PA28γ, there are distinctive details leading to significantly divergent functions that regulate 20S activity. 

### 6.2. Physiology

Arguably, the existing literature offers a more comprehensive exploration of PA28γ’s role in physiology and disease as compared to its biochemical properties. This is largely because PA28γ was initially discovered during research studies into the serum of patients suffering from systemic lupus erythematosus (SLE). From its discovery, PA28γ has been implicated in many physiologic processes, such as cell proliferation, apoptosis [86,87,88], lipid metabolism [89], DNA repair [90,91], fertility [92], nuclear and chromatin organization [93], and a plethora of diseases including: cancers (breast [94,95], thyroid [96], lung, skin, colorectal [97,98,99], and oral [100]), neurodegeneration [101,102,103], cardiac hypertrophy [104], Hepatitis C infection [89,105], and SARS-CoV-2 [106].

Interestingly, experimental overexpression of PA28γ in poly-glutamine (Poly-Q) neurodegenerative disease (e.g., Huntingdon’s Disease) and brain disorder [102] models demonstrates an increase in cell survival and amelioration of the motor neurodegenerative disease phenotypes. For a more in-depth discussion of PA28γ’s role in physiology and disease, see Cascio, 2021 [107]. Moreover, omics-based analyses have detected an age-related decrease in the expression of this protein, with a substantial decline also identified in patients suffering from Alzheimer’s disease, interestingly correlating with Tau protein levels [108,109,110,111]. This was recently corroborated by Tu, 2022, where it was demonstrated that the PA28γ–20 CP complex mediates the ubiquitin-independent degradation of phosphorylated nuclear Tau [111]. A deficiency in this complex promotes neurodegeneration, while its proper function regulates the accumulation of this key protein in Alzheimer’s disease in vivo, hence relieving tauopathy-related cognitive impairments in murine models [111]. This discovery opens a new avenue in the regulation of tau and links directly to PA28γ as one candidate for the designing of new therapies.

Under normal conditions, the role of PA28γ has been shown to boost proteostasis/survival in cold environments, as evidenced by recent studies on *C. elegans* and mammalian cell lines [112]. The RNA silencing of this protein markedly reduces the lifespan of organisms subjected to moderate cold environments and severely abolishes their ability to degrade PolyQ aggregates and other aging-associated pathological protein species that resist proper degradation via ubiquitination pathways, such as IFB2. It is also suggested by Lee, 2023, that PA28γ-mediated degradation is not solely confined to the nuclear region, as traditionally expected [112]. This finding, establishing PA28γ’s presence in various cellular compartments, indicates potential unexplored roles in modulating the proteostasis network and aging processes under non-pathological conditions. Intriguingly, the functionality of PA28γ seems to have been preserved throughout evolution and is linked to animal proteostasis and thermal response [112]. This makes the study of PA28γ attractive for the discovery of new therapeutic targets or activators to treat proteinopathies, and increases the rising interest in cold temperature therapies and proteasome-facilitated degradation of pathological protein aggregates.

Despite advancements in understanding the role of PA28γ in these diseases and its impact on disease progression, we still know relatively little about the mechanism and the partners involved in contributing to the observed phenotypes. This knowledge gap can be largely attributed to our incomplete understanding of PA28γ’s 20S CP regulatory mechanism and the scarce information about the specific proteins that the PA28γ–20S CP complex is responsible for degrading, or its substrate-selection mechanisms. This gap in knowledge also raises questions about the causal relationship between PA28γ dysregulation and disease: does a dysregulation in PA28γ expression precipitate disease, or does the progression of the disease itself prompt changes in PA28γ expression?

The contrast between the highly specialized role that inducible PA28αβ serves in physiology and the broad spectrum of physiological processes and disease implications associated with the constitutively expressed PA28γ is fascinating. Particularly unique to PA28γ is the range of proteins it interacts with and its demonstrated ability to facilitate their degradation. Some remarkable examples include PA28γ regulation in the degradation of p53 [113], p21 [114], SRC-3/AIB1 [115], Smurf1 [116], Oct-1 [117], Hepatitis C virus core protein [89], and SARS-CoV-2 nucleocapsid protein [106]. Although proteins participate in distinct cellular functions, they share extensive unstructured regions, which have recently been demonstrated to be susceptible to degradation through the PA28γ–20S CP complex [48]. These findings, in conjunction with the established role of PA28γ as a 20S CP T-L activator, which also has a gate-opening capacity, prompts a range of novel and exciting questions about the physiological impact of PA28γ, extending beyond nuclear proteostasis to encompass overall organismal homeostasis.

## 7. Concluding Remarks

Significant effort has been dedicated to the biochemical and physiological characterization of the mammalian 11S family of proteasomal regulators. The understanding of PA28αβ has greatly evolved since its identification as a gate-opener of the 20S CP, particularly in comprehending its role in MHC Class I antigen presentation and its regulation of immunoproteasome peptide product size [56]. Moreover, comprehensive structural characterization of both the PA28αβ [38] and PA28αβ-immunoproteasome [72] complexes has elucidated its biochemical function and expanded our perspective, especially in comparison with other eukaryotic 11S family members.

Similar progress has been made regarding the other mammalian 11S homolog, PA28γ. Recent research demonstrates that PA28γ can degrade unfolded proteins [48], reinforcing PA28γ’s capability to facilitate the 20S CP degradation of specific proteins mentioned above. Nevertheless, the precise mechanism by which this complex facilitates the degradation of these proteins remains elusive. Conflicting biochemical evidence of PA28γ stimulating the 20S CP’s T-L catalytic site or gate-opening complicates our ability to understand its biochemical mechanism of unfolded protein degradation and its role in physiology and disease progression. This scenario inevitably raises the question: is it possible for this regulator to execute both functions in an in vivo context? Especially considering the 20S CP’s difficulties in degrading polyQ expanded repeat proteins that are highly charged [118], and the fact that the T-L site, with its function of cleaving post-glutamine, is still the least processive among the proteasomal catalytic sites [45,119], the question becomes even more interesting. Could PA28γ’s ability to upregulate T-L activity enhance polyQ protein degradation? Could this mechanism be targeted by small molecules? Achievement of a high-resolution PA28γ–20S CP complex structure would enable visualization and the precise 20S CP conformational changes when in complex with PA28γ, which should provide evidence of its precise biochemical mechanism and boost the future development of small-molecule drugs that could activate the T-L site. 

Elucidating this molecular mechanism and the diverse range of binding partners is essential to establish a robust basis to better understand the role PA28γ plays in physiology, and why this complex governs the degradation of certain proteins, especially unstructured proteins. Moreover, this foundational knowledge will facilitate connections between homeostatic mechanisms and PA28γ’s role in pathophysiology and provide a novel approach for the development of targeted therapeutics against PA28γ in disease.

## Figures and Tables

**Figure 1 biomolecules-13-01326-f001:**
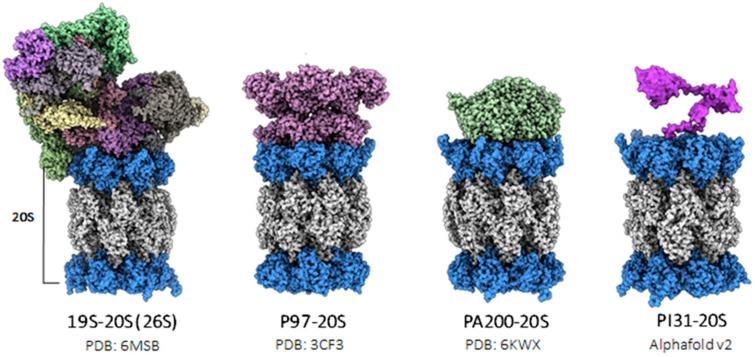
Structures of common proteasomal regulators. Solved structures of proteasomal regulators: the 19S–20S (26S) complex, extensively studied, and the PA200–20S complex. The P97 structure (PDB: 3CF3) is placed on top of the H20S structure (PDB: 6MSB). PI31, predicted by AlphaFold v2.0, is imposed on the H20S; this structure configuration has not been resolved. The C-terminal domain of PI31, observed within the interior of the 20S, is illustrated in PDB: 8FZ6. The 19S and P97 are members of the AAA+ ATPase family. P97 functions as an unfoldase that feeds unfolded substrates into the 20S proteasome, while PA200 and PI31 are ATP-independent regulators, employing unique strategies to modulate the 20S.

**Figure 2 biomolecules-13-01326-f002:**
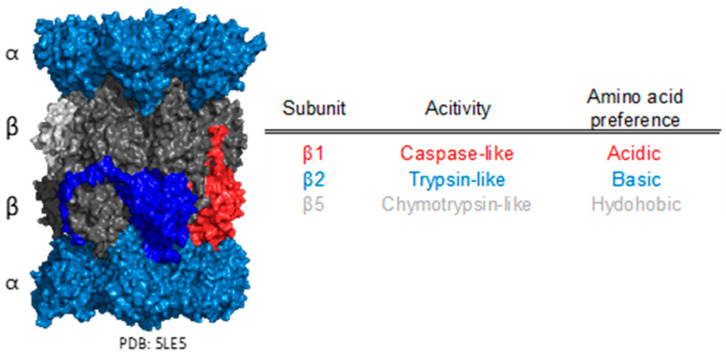
Eukaryotic 20S CP structure and catalytic function. 20S CP has four heteroheptameric rings, two α (blue) and two β (dark gray), arranged in an α, β, β, α arrangement. The α-rings act as a gate to deter non-specific substrate entry. The β-rings have three subunits that contain proteolytic activity: β1-caspase-like (red), β2-trypsin-like (royal blue), and β5-chymotrypsin-like (light gray).

**Figure 3 biomolecules-13-01326-f003:**
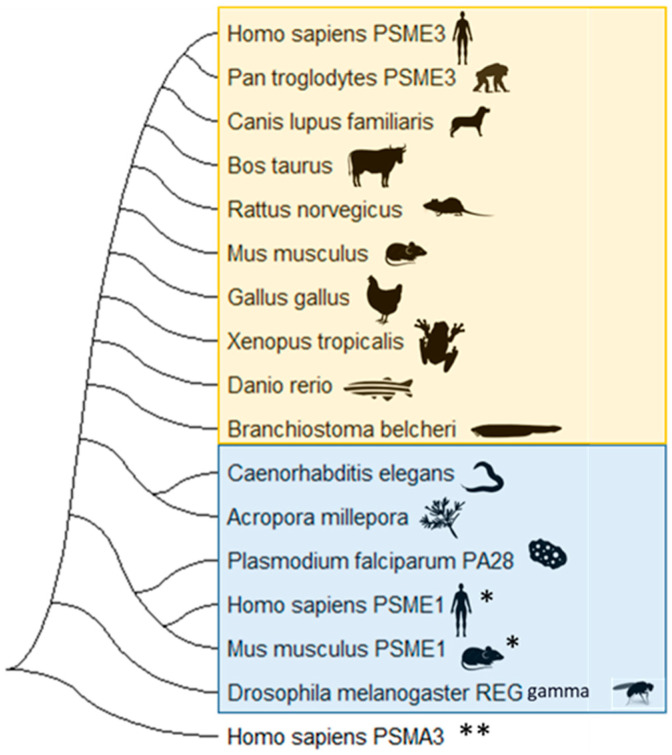
Evolutionary relationships of PA28γ. The phylogenetic tree in this figure represents a comparative analysis of PA28γ sequences in representative organisms obtained from the NCBI database. The orange shade presents the cluster obtained for the vertebrate PA28γ (a highly conserved PA28γ cluster in vertebrates). In contrast, the blue shade highlights the invertebrate PA28/REG gamma variants. These variants cluster closely with PA28α (denoted by sequences with *), emphasizing their closer sequence similarity to vertebrate PA28α as opposed to vertebrate PA28γ. This pattern hints at a possible diversification of this protein family during vertebrate evolution. For this tree, the relationships (evolutionary history) were inferred using the NeighborJoining method. The bootstrap consensus tree inferred from 5000 replicates is taken to represent the evolutionary history of the species analyzed. Branches corresponding to partitions reproduced in less than 50% of bootstrap replicates are collapsed. The evolutionary distances were computed using the Poisson correction method and are in the units of the number of amino acid substitutions per site. This analysis involved 17 amino acid sequences. All ambiguous positions were removed for each sequence pair (pairwise deletion option). There was a total of 313 positions in the final dataset. Evolutionary analyses were conducted in MEGA11. Note: sequences marked with * represent sequences from vertebrate PA28α. The alpha subunit of the proteasome (PSMA3 **) was utilized as an outgroup to root the tree.

**Figure 4 biomolecules-13-01326-f004:**
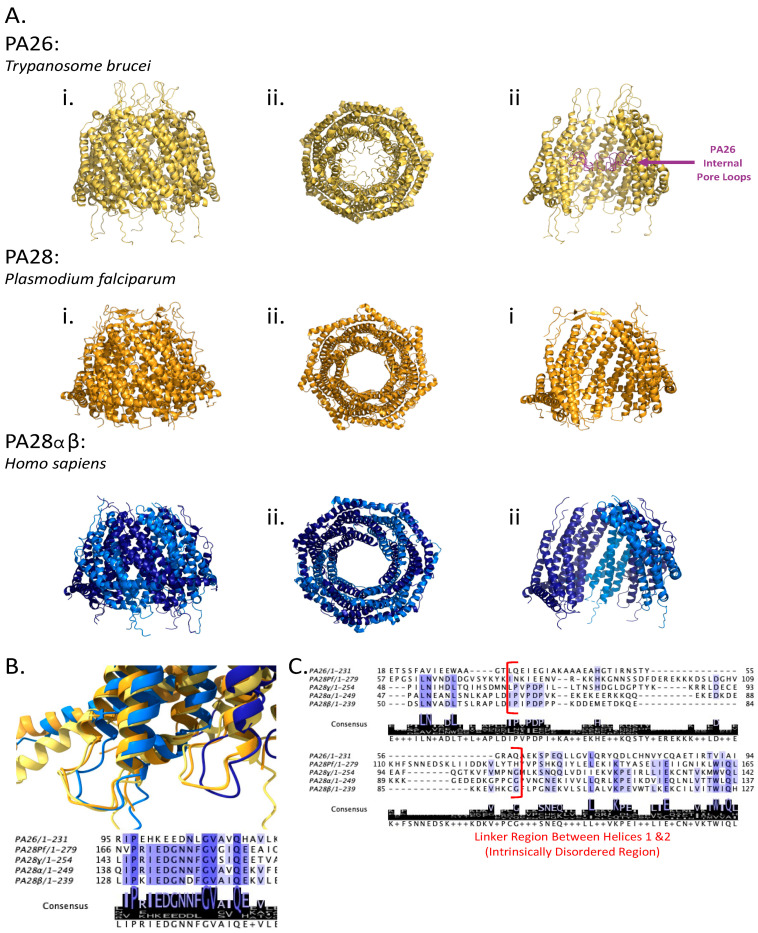
Structural comparison of 11S regulators. (**A**) **i.** side, **ii.** top-down, and **iii.** sliced views of PA26 (yellow, with internal pore loops as purple; PDB: 1Z7Q), PA28Pf (orange; PDB: 6DFK), and PA28αβ (α as dark blue; β as light blue; PDB: 7DR6). (**B**) Overlay of PA26, PA28Pf, and PA28αβ’s activation loops. Aligned sequences are below and conserved residues are highlighted from least conserved (white) to highly conserved (dark purple). (**C**) Sequence alignment of the 11S family members. Bracketed (red) is the IDR between helices 1 and 2.

**Figure 5 biomolecules-13-01326-f005:**
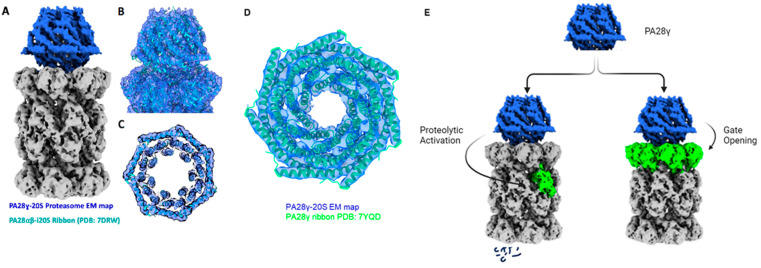
PA28γ structural comparisons with the PA28αβ–i20S complex and PA28γ Atomic Model. (**A**) side view of the PA28γ (blue)–20S CP (gray) complex electron density after Denmod, (**B**) PA28αβ–i20S CP model fitted into the electron density of PA28γ–20S CP, and (**C**) top-down view of density fit, highlighting the fit of PA28αβ’s secondary structures into the electron density of PA28γ. (**D**) density fit of PA28γ secondary structures into the electron density of PA28γ, presented by Thomas, 2022. (**E**) potential regulatory mechanisms underlying PA28 λ function.

## Data Availability

Not applicable.

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
