# Peer review of "Structure, Function, and Allosteric Regulation of the 20S Proteasome by the 11S/PA28 Family of Proteasome Activators"

_biomolecules, 2023, doi:10.3390/biom13091326_

Round 1
Reviewer 1 Report
This is a well thought out thesis on a timely topic. The main author, an expert on the association of ATPases with the 20S chamber, uses his knowledge to bring forth new insight into the association of other N=non-ATPases regulators of the 20S. The review is well-written and will serve as a go-to source of knowledge for the next few years.
biomolecules-2537355 is a timely review of a topic that has been known for about 25 years but is carefully summarized and updated with recent findings. The authors bring fresh insight into the similarities and divergences of different 11S regulators of the 20S proteasome and the evolutionary links between the alpha, beta, and gamma variants of 11S reg / PA28. Minor comment: The legend of Fig 3 is a bit confusing and possibly contains a few typos. Is the lowest branch PSMA3 (caption) or PSMEA3 (legend) or PSME3? Likewise the blue vs orange shading does not seem consistent (shouldn’t Reg gamma be in the orange box?!?). Please read carefully and clarify the legend and corresponding text in rows 250-264.
Author Response
Find attached.

Reviewer 2 Report
The review by Thomas et al. focusses on the role and function of the family of proteasome activators PA28. Overall, this review is clear and well written and provides interesting recapitulation of how PA28 interacts with the 20S proteasome. However, reading the manuscript, I feel that the part related to the PA28ab regulator is less clear and detailed than the part relating to PA28gamma. Here are a few comments to ameliorate the quality of the manuscript:
First, a description of what is immunoproteasome and how it is induced should be introduced earlier in the “Proteasome structure and function paragraph”. The term “Immunoproteasome” is first cited in line 254 (with no explanation), but the nature of this proteasome complex is only provided at a later stage (line 372).
The role of PA28ab in MHC I presentation is actually not that clear and still controversial. Its role on the actual production of MHC I antigens is only shown for a few specific peptides. I feel that the authors should be more cautious when claiming that PA28ab is required for MHC I presentation (in particular lines 222, 368 and 379) and they should use references other than review articles when clarifying this point.
Regarding the role of PA28ab in the degradation of oxidized proteins, a more careful description of the relevant observations on this matter should be done, citing the original papers.
Minor comments:
Line12 : add an s at disease : different types of diseases
Line69: replace “of” by “by”: degradation by the proteasome
Line 426: remove In and it : 2007). PA28ab has been suggested …
Author Response
Find Attached

Reviewer 3 Report
The manuscript by Thomas and co-workers is a summary of the known literature about the role of PA28 in proteasomal degradation. The authors provide an overview of proteasome regulation, focusing on regulators of ubiquitin-independent pathways of the 11S family of regulators. In particular, the authors highlighted the evolutionary conservation of PA28λ, its structure-function relation with the proteasome core, and its physiological roles. I find the manuscript an important summary of recent data, including PA28λ's possible mechanisms of function and roles in health and disease states. I only have a few comments:
- The authors should go through the entire manuscript and correct ambiguities and cumbersome text. For example, Line 91: "The 20S by itself, PA200-20S, and PA28-20S, complexes have all been seen to catalyze unstructured protein degradation. These degradation pathways are also considered particularly important to health and disease". Since these are published data, I favor a more direct writing, such as "complexes can catalyze…" "These degradation pathways are particularly important…". Line 160, " and later demonstrated more conclusively" unclear what the authors refer to. Line 220, instead of what? Line 421, "These hypothetical mechanisms for PA28γ find their foundation in the established functions of other 20S CP regulators". I am not sure what was the authors' intention.
- Line 276, I don’t understand the argument. I do not think it’s a valid question that the authors can answer.
- Quality of images seems to be low. Are the final figures of better quality?
- Since the review is intended to attract the attention of a broad audience, it would be advantageous to add an explanatory figure illustrating the possible mechanisms of PA28 λ function.
- Line 31: PTMs. It is an abbreviation that should be clarified.
Described above
Author Response
Find attached
